# Explosive Nucleosynthesis Study Using Laser Driven γ-ray Pulses

**Takehito Hayakawa [1,2,\*], Tatsufumi Nakamura [3], Hideyuki Kotaki [4], Masaki Kando [4] and Toshitaka Kajino [2,5,6]**

[1]   National Institutes for Quantum and Radiological Science and Technology, 2-4 Shirakata, Tokai, Naka, Ibaraki 319-1106, Japan

[2]   National Astronomical Observatory of Japan, Mitaka, Tokyo 181-8588, Japan; kajino@nao.ac.jp

[3]   Fukuoka Institute of Technology, Fukuoka, Fukuoka 811-0295, Japan; t-nakamura@fit.ac.jp

[4]   National Institutes for Quantum and Radiological Science and Technology, Kizugawa, Kyoto 619-0215, Japan; kotaki.hideyuki@qst.go.jp (H.K.); kando.masaki@qst.go.jp (M.K.)

[5]   The University of Tokyo, Bunkyo-ku, Tokyo 113-0033, Japan

[6]   Beihang University, Beijing 100191, China

\*   Correspondence: hayakawa.takehito@qst.go.jp; Tel.: +81-70-3943-3386

**Abstract:** We propose nuclear experiments using γ-ray pulses provided from high field plasma generated by high peak power laser. These γ-ray pulses have the excellent features of extremely short pulse, high intensity, and continuous energy distribution. These features are suitable for the study of explosive nucleosyntheses in novae and supernovae, such as the γ process and ν process. We discuss how to generate suitable γ-ray pulses and the nuclear astrophysics involved.

**Keywords:** nuclear photoreaction; supernova nucleosynthesis; γ process; ν process; p process

## 1. Introduction

Progress in laser physics has enabled us to produce various quantum beams, such as electrons, gamma-rays, and ion beams from high field plasma generated from the interaction between high peak power laser and materials [1]. Because the energies of these radiations can be higher than 10 MeV, they can interact with atomic nuclei as well as atoms. Thus, they have the potential to be used for the study of nuclear physics. These laser-driven quantum beams have the following features: high flux, ultra-short pulse in a range from femtosecond to nanosecond, and continuous energy distribution. These features are particularly suitable for the study of stellar nucleosyntheses in nuclear astrophysics. About 99% of elements heavier than iron are considered to be synthesized by the two major processes of rapid neutron capture reactions (r process) and slow neutron capture reactions (s process). In the Extreme Light Infrastructure Nuclear Physics (ELI-NP), it was proposed to produce extremely neutron rich unstable isotopes using fission–fusion reactions on multi-targets including $^{232}$Th with a high peak power laser to study the r process [2]. It was also proposed to study the s process and photodisintegration reactions in supernova explosions (γ process) using a highly intense neutron pulse provided from laser driven D-T nuclear fusion reactions at the Nuclear Ignition Facility (NIF) in Lawrence Livermore National Laboratory [3]. A pioneering laser experiment for the Big-Bang nucleosynthesis which played a significantly important role for syntheses of light elements at the beginning of the universe was performed at University of Texas at Austin [4]; the astrophysical cross-section of the $^3$He(d, p)$^4$He reaction in the plasma generated by the Texas Petawatt Laser was measured.

In the energy region of E > 10 MeV, nuclear reactions inside of plasma generated by high intensity laser strongly affect the generation of quantum beams. Thus, Nakamura and Hayakawa [5] developed

a particle-in-cell (PIC) simulation code considering nuclear reactions including photodisintegration reactions. Using this code, it was presented that various radiations, such as $\gamma$ rays, neutrons, ions, electrons, and positrons could be generated [6]. We focus our attention on nuclear photoreactions with continuous energy $\gamma$-ray beam generated by high peak power laser to study nuclear astrophysics. Photons in the MeV energy region play an important role in explosive nucleosyntheses in novae and supernovae such as $\gamma$ process and neutrino-induced reactions in supernova explosions ($\nu$ process). Continuous energy $\gamma$-rays have been generated using high peak power lasers, although their measurements have been a technical challenge (for example, see [7–9]). In this paper, we propose nuclear experiments using laser-driven $\gamma$-ray pulses to simulate stellar nuclear photoreactions. We explain the astrophysical background for the proposed experiments. We present new concepts of three types of experiments using laser-driven $\gamma$ pulses with stellar energy distribution: (i) the direct measurement of the stellar ($\gamma$, n) reaction cross section on the ground state of atomic nuclei; (ii) the direct measurement of the ($\gamma$, n) cross section on excited states; and (iii) the direct measurement of the transition probability between the ground state and an isomer of astrophysical interest via ($\gamma$, $\gamma'$) reactions. An important point to consider in these methods is the generation of $\gamma$-ray pulse with energy distribution approximately identical with stellar photon distribution, and thus we discuss how to generate $\gamma$ pulses suitable for the currently proposed experiments.

## 2. Calculation Method and Results

It is highly desirable in all proposed methods to consider how to generate the qusai-stellar energy $\gamma$-ray pulse. Thus, we demonstrated the generation of $\gamma$-ray pulses using a particle-in-cell (PIC) code. In a previous work [5], Nakamura and Hayakawa developed a PIC code which includes the generation and transport of $\gamma$-rays in targets in order to explore the energy transport in laser–matter interactions. In this code, as photon transport processes, Compton scattering, electron–positron pair creations under the nuclear electric field, and photonuclear reactions of ($\gamma$, p), ($\gamma$, n), ($\gamma$, 2n), and ($\gamma$, $\alpha$) were included. These processes can be treated as a Monte-Carlo manner as used in particle transport codes. First, using this code [5], we calculated a $\gamma$-ray energy spectrum of plasma generated in a solid carbon target with a highly intense laser with a wavelength of 0.8 $\mu$m, an energy of 300 J, a peak power of 10 PW, and duration of 30 fs. The intensity on the target was $5 \times 10^{22}$ W/cm$^2$, the normalized amplitude of which was a = 150. We assumed the carbon foil with a thickness of 2 g/cm$^3$. Next, we calculated a $\gamma$-ray spectrum generated by bremsstrahlung of continuous energy electrons accelerated by the laser plasma using a Particle and Heavy Ion Transport Code System (PHITS) particle transport simulation code [10]. Previous studies reported that electron beams with continuous energies and temperature of several hundred keV could be generated by the laser plasma interactions [11,12]. We assumed that electrons with an energy of $kT$ = 300 keV (where $k$ is Boltzmann constant) was generated from the main target by laser plasma interactions and that subsequently the $\gamma$-rays were generated by bremsstrahlung on the solid gold target located behind the main target.

Figure 1 shows a calculated example of a $\gamma$-ray spectrum generated predominantly by the radiation effect. Above $10^7$ eV, the $\gamma$-ray flux decreases as photon energy increases, but the peak locates at 1–3 MeV. This suggests that $\gamma$-rays with energies of 1–10 MeV are effectively generated via radiation reaction effects. Figure 2 shows an example of the energy spectrum generated by the bremsstrahlung on the solid gold target, in which the primary electron beam was generated by the laser-induced plasma. The maximum energy of the $\gamma$-rays is approximately 10 MeV. The red line shows the Planck distribution of an energy of $kT$ = 225 keV. These two lines clearly show that the calculated energy spectrum can be reproduced by the Planck distribution—namely the stellar photon spectrum—with temperature of $10^8$–$3 \times 10^9$ K.

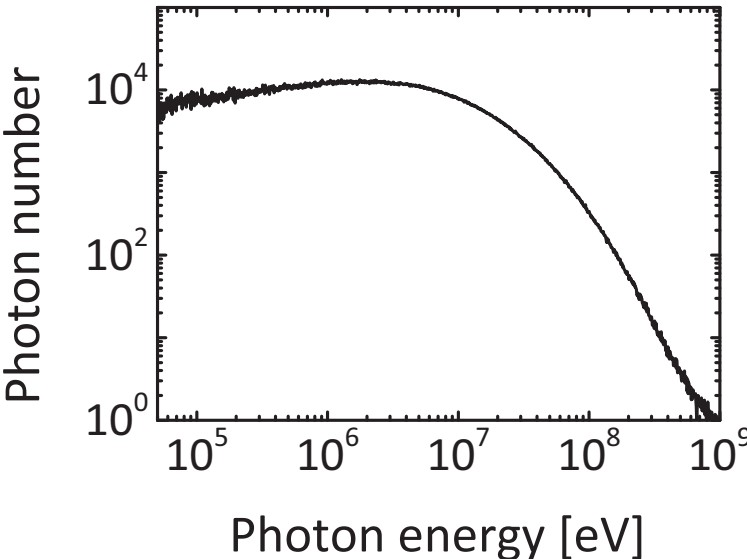

**Figure 1.** An example of the energy spectrum of $\gamma$-rays generated predominantly by the radiation effect. Above $10^7$ eV, the $\gamma$-ray flux decreases as photon energy increases, whereas the peak is located at 1–3 MeV.

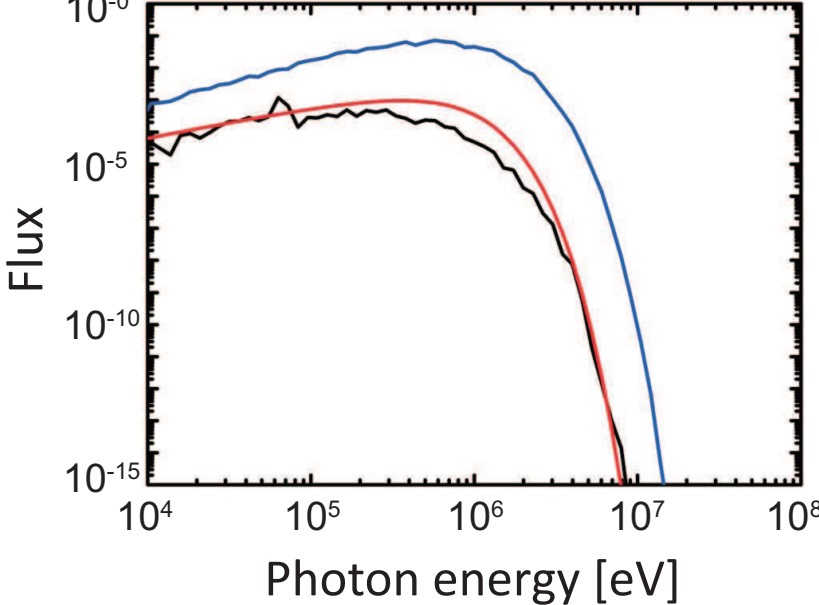

**Figure 2.** The calculated energy spectrum of the $\gamma$-rays generated by the bremsstrahlung on the solid gold target. The photon flux suddenly decreases above 1 MeV. The blue line indicates the energy spectrum of the primary electron pulse with energy of $kT = 300$ keV. The red line shows the Planck distribution with energy of $kT = 225$ keV. The bremsstrahlung spectrum can be reproduced by the Planck distribution.

## 3. Discussion

### 3.1. Role of Photons in Explosive Nucleosynthesis

Massive stars that are heavier than one solar mass by greater than or equal to a factor of eight culminate their evolution as core-collapse supernova explosions. At the early phase of supernovae, a huge number of neutrinos are emitted from the proto-neutron star in the center of the massive

star (see Figure 3). Most neutrinos pass through outer layers of the star, whereas some fractions of neutrinos deposit their energy in outer layers through the interaction with materials. The heating by this deposited energy leads to the final explosion. High-energy photons at energies of a MeV or more are created in the high-temperature environments in outer layers, and these photons can synthesize new isotopes—so-called "p nuclei"—by successive photon-induced reactions such as ($\gamma$, n), ($\gamma$, p), and ($\gamma$, $\alpha$) reactions ($\gamma$ process). Historically, 35 isotopes are classified into the p nuclei. This $\gamma$ process was proposed by Woosley et al. in 1978 [13] and recent studies show that 27 p-nuclei were synthesized by the $\gamma$ process [14,15]. The p nuclei are characterized by being at the neutron-deficient extremes of the stable nuclides and being generally of very low abundance relative to the other isotopes of each element; typically their isotopic abundance ratio are lower than 1%. Figure 4 shows a partial nuclear chart, and nucleosynthesis flows around La and Ce nuclides. $^{136,138}$Ce are the p nuclei, which can be produced by successive ($\gamma$, n) reactions from heavy isotopes such as $^{140}$Ce [14]. The synthesized p nuclei are also destroyed by photon-induced reactions, as shown in Figure 4. In this way, high energy photons contribute to both of the production and the destruction of p nuclei.

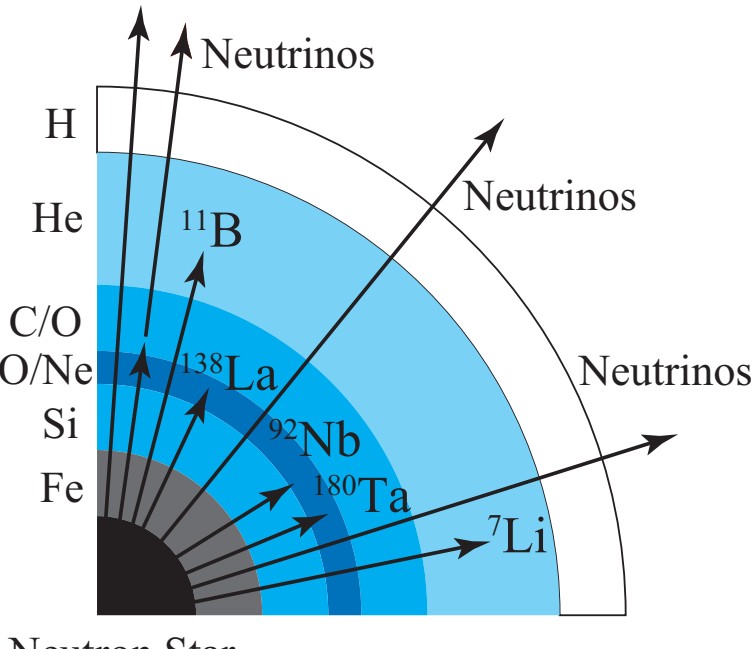

**Figure 3.** Schematic view of neutrino-process in core-collapse supernova explosions. At the early phase of the supernova explosion, a huge number of neutrinos are emitted from the proto neutron star. Most neutrinos escape without any interactions with materials. Some fractions of neutrinos deposit their energy in outer layers. In O/Ne layers, p nuclei are synthesized by the $\gamma$ process. Some $\nu$ isotopes are also produced by the $\nu$ process.

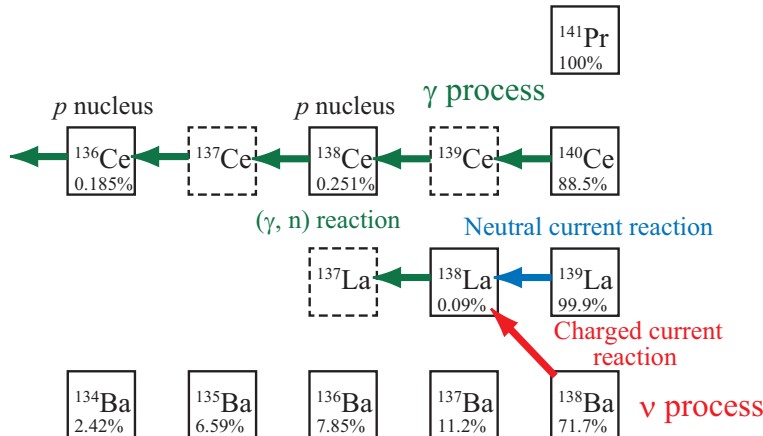

**Figure 4.** Partial nuclear chart and nucleosynthesis flows around Ce, La, and Ba. $^{138}$La is a $\nu$ isotope which is produced predominantly by the charged current reaction and neutral current reaction in the $\nu$ process, whereas $^{138}$La is also destroyed by the ($\gamma$, n) reaction. $^{136}$Ce and $^{138}$Ce are p nuclei which are predominantly produced by the successive ($\gamma$, n) reactions from heavy isotopes such as $^{140}$Ce.

Neutrinos can generate new isotopes by nuclear reactions on pre-existing nuclei in outer layers. Because the neutrino–nucleus interaction is extremely weak, the neutrino process alone can play a significant role in the syntheses of isotopes which are bypassed by other major processes such as r, s, and $\gamma$ processes. The neutrino process in supernova explosions was proposed as the astrophysical origin of rare isotopes such as $^7$Li, $^{11}$B, $^{19}$F, $^{138}$La, and $^{180}$Ta [16,17]. Among elements heavier than iron, only two isotopes of $^{138}$La and $^{180}$Ta were known as the $\nu$ isotopes. In 2013, Hayakawa et al. [18] proposed the $\nu$-process origin for $^{92}$Nb, which is a $\beta$-unstable isotope with a half-life of $3.5 \times 10^7$ y. $^{92}$Nb does not exist at the present solar system, but it was found that the evidence that $^{92}$Nb existed at the time of the solar system formation in primitive meteorites analyses, which show an anomaly in $^{92}$Zr abundance. The initial abundance of $^{92}$Nb at the solar system formation was evaluated, but the astrophysical origin of $^{92}$Nb is an unresolved problem. Hayakawa et al. [18] presented that the initial abundance of $^{92}$Nb could be explained by a scenario wherein a supernova explosion happened at the time $10^6$–$3 \times 10^7$ y. before the solar system's formation. Note that the origin of $^{92}$Nb is still a subject of study [19,20]. In this way, the study of nucleosyntheses can contribute to the understanding of the history before the solar system's formation as well as the origin of materials. The $\nu$ isotopes are produced by neutrino-induced reactions, but newly synthesized nuclei can be destroyed predominantly by photodisintegration reactions like the $\gamma$ process. For example, a neutrino isotope $^{138}$La is predominantly produced by the two neutrino induced reactions of the charged current reaction of $^{138}$Ba($\nu_e$, e$^-$)$^{138}$La and the neutral current reaction of $^{139}$La($\nu$, $\nu'$n)$^{138}$La (see Figure 4). The synthesized $^{138}$La may be destroyed by the ($\gamma$, n) reaction. Thus, high energy $\gamma$-rays play the important role in the $\nu$ process.

### 3.2. Direct Measurement of Stellar Nuclear Reaction Rate on the Ground State

In this paper, we propose three experimental methods for the study of stellar nucleosynthesis using laser-driven $\gamma$-ray pulses with an energy spectrum approximately equal to the stellar Planck distribution. First, we explain the direct measurement of stellar nuclear reaction rates on the ground state. Experimentally measured rates for photon-induced reactions are essential for accurate theoretical simulations of the $\gamma$ and $\nu$ processes. In O/Ne rich layers, where the p nuclei are synthesized by the $\gamma$ process, the peak temperature at supernova explosions increases to $T \cong (1$–$3) \times 10^9$ K. The energy distribution of a thermal photon bath at a temperature $T$ is presented by the Planck distribution:

$$n(E, T) = \left(\frac{1}{\pi}\right)^2 \left(\frac{1}{\hbar c}\right)^3 \frac{E^2}{exp(E/kT) - 1},$$ (1)

where $n(E, T)$ is the number of photons at energy $E$ per unit of volume and energy interval, $\hbar$ is Dirac constant, and $c$ is the speed of light (see the filled area in Figure 5). The temperature-dependent nuclear reaction rate $\lambda(T)$ is given by

$$\lambda(T) = \int_0^\infty cn(E, T)\sigma(E)dE. \tag{2}$$

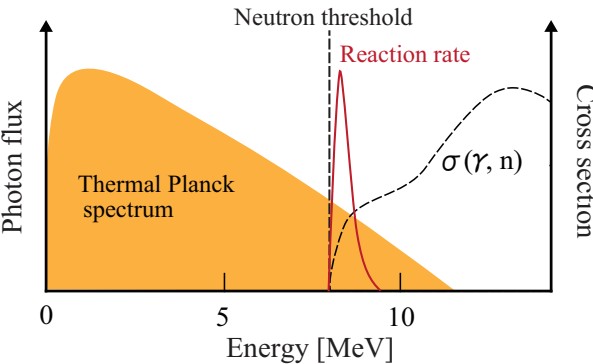

**Figure 5.** Schematic view of the energy spectrum of photons, photon-induced reaction cross-sections, and reaction rate in hot temperature environments. The filled area shows a typical Planck distribution of photons. The dashed-line shows a typical ($\gamma$, n) reaction cross section. The *Y*-axis is presented in log scale. The solid line indicates the reaction rate, which simply means flux × cross-section. The concept of this schematic view was adapted from Mohr et al. [21].

The ($\gamma$, n) reactions can be understood by a two-step model (see Figure 6). First, a nucleus can be excited by the absorption of a photon. Next, if the excitation energy of the nucleus is higher than the neutron threshold (which is typically 7–10 MeV), the excited state can decay to a residual nucleus by the emission of a neutron. The distribution of photons in supernovae is Planckian as discussed above, so the photon flux sharply decreases with increasing photon energy. Thus, the energy at which the photons contribute to the abundances of the p nuclei is typically a few hundred keV above the neutron threshold (see the solid line in Figure 5).

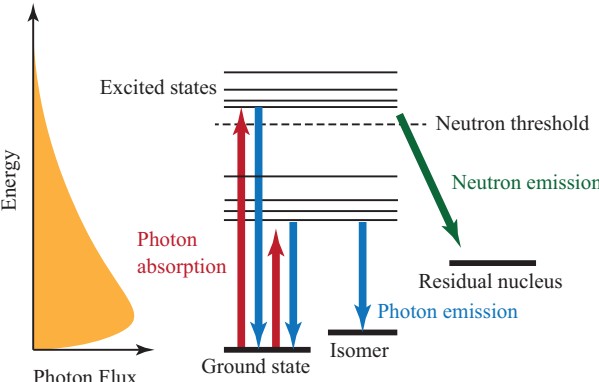

**Figure 6.** Schematic view of the interaction between an atomic nucleus and a photon. The nucleus is excited by the absorption of a photon. The excited state can decay to the ground state or the isomer by the emission of a photon (photons). If the excitation energy is higher than the neutron threshold, the state can decay to the residual nucleus by the emission of a neutron.

The nuclear reaction cross-section has been measured using quantum beams provided from various accelerators. These quantum beams (except for bremsstrahlung) have a feature of monochromatic energy, $\delta E/E < 10^{-3}$, where $E$ and $\delta E$ are the average energy of the quantum beam and the energy spread, respectively. The stellar reaction rate as a function of temperature

is obtained by integrating the cross-section measured using monochromatic energy beams. A method to directly measure the stellar reaction rate has been used for the study of the s process, in which neutron beams of stellar continuous energies at $kT$ = 25–30 keV were generated by the $^7$Li(p, n) reaction with proton beams provided from accelerators [22,23]. The thickness of $^7$Li targets and the scattered angles of neutrons have been tuned to obtain the neutron beam with the quasi-stellar energy spectrum. The $^3$H(p, n) and $^{18}$O(p, n) reactions are also used for $kT$ = 52 keV [24] and 5 keV [25], respectively. For the study of the $\gamma$ process, Mohr et al. proposed a similar method using bremsstrahlung $\gamma$-rays [21,26], although the energy distribution of the bremsstrahlung $\gamma$-ray differs from the Planck distribution. To simulate the stellar Planck distribution, they measured the reaction rate for different maximum energy bremsstrahlung $\gamma$-rays and integrated them after the measurements. Because this method is not easy, few nuclides were studied using this method. At present, the cross-sections of photodisintegration reactions on seed isotopes heavier than the p isotopes in the same element have been systematically measured using laser Compton scattering $\gamma$-ray beam with a qusai-monochromatic energy [27]. However, the ($\gamma$, n) reaction cross-sections on the p nuclei have not been well measured.

A feature of the laser-driven $\gamma$-ray pulse is continuous energy distribution. Therefore, we propose the direct measurement of a stellar nuclear reaction rate using a laser-driven $\gamma$-pulse with energy spectrum approximately equal to the Planck distribution at the astrophysical energy region. Standard cross-section measurement methods such as the activation method and the time-of-flight method can be applied to the present prosed method. An important consideration in this method is how to generate quasi-stellar energy distribution. A candidate is the $\gamma$-rays generated by the radiation effect [28]. High energy particles generated by laser plasma interactions should be decelerated inside of the plasma, and thereby high energy $\gamma$-rays can be produced by synchrotron radiations following the decelerations of the particles. Figure 1 shows that the $\gamma$-rays in the MeV energy region can be effectively generated by the radiation effect. It is well known that the $\gamma$ rays can be generated by the bremsstrahlung with electrons generated by laser-plasmas [29]. If an electron beam with well-tuned continuous energies will be generated, its bremsstrahlung $\gamma$-rays may have quasi-stellar energy distribution as shown in Figure 2.

*3.3. Direct Measurements of Nuclear Reactions on Excited States*

Next, we discuss the direct measurements of nuclear reactions on excited states using a laser-driven $\gamma$-pulse. In the previous subsection, we discussed direct measurements of the cross-section on the ground state. However, nuclei are thermalized in the high-temperature regime, and hence we should consider nuclear photoreactions on excited states. The population ratio of any two states in thermalized nuclei is simply given by a quotient of their Boltzmann factors:

$$\frac{m_i}{m_j} = \frac{2J_i + 1}{2J_j + 1} exp \left( \frac{-(E_i - E_j)}{kT} \right), \tag{3}$$

where $m_i$ denotes the population of the state $i$ with spin $J_i$ and excitation energy $E_i$. The normalized population ratio of the excited state of $i$ to the summation of all states in a nucleus, $P_i$, is given by

$$P_i = (m_i/m_0) / \sum_j (m_j/m_0), \tag{4}$$

where the notation of 0 indicates the ground state. In such conditions, the nuclear reaction rate $\lambda(T)$ should be modified to the stellar reaction rate

$$\lambda^*(T) = \sum_i P_i \lambda_i(T), \tag{5}$$

where $\lambda_i(T)$ is the partial reaction rate of the state $i$. The required energy for the ($\gamma$, n) reaction on an excited nucleus is reduced by its excitation energy from the neutron threshold. This means that the stellar reaction cross-section suddenly increases in hot temperature such as $\gamma$-process layers in supernova explosions. For the study of the $\gamma$-process, the stellar reaction rates on thermalized nuclei have been evaluated using theoretical calculation with the cross-section on the ground state. Vogt et al. [26] measured the cross-section on the ground states of Pt isotopes and calculated the cross-sections on thermally excited nuclei; it was pointed out that the calculated cross-sections at temperatures of $T = (2\text{--}3) \times 10^9$ K were greater than the measured cross sections on the ground states by two-to-four orders of magnitude. These huge enhancements suggest the need for the direct measurement of the stellar reaction rates.

Because the half-lives of excited states in nuclei are too short to interact with a photon before its decay, the direct measurement of $\lambda_i(T)$ of excited states is extremely difficult, although various indirect methods such as inverse reactions were developed. An advantage of the laser-driven quantum beam is its extremely short pulse. The nuclei in the plasma can be excited by the absorption of the plasma radiation and by various atomic-nucleus interactions such as nuclear excitation by electron transition (NEET) [30]. If an excited state which was previously populated by laser plasma is irradiated by high intensity radiation pulse, the reaction rate of the excited state, $\lambda_i(T)$, can be directly measured. The half-lives of excited states with excitation energies of $0-500$ keV are typically in the range from picosecond to nanosecond. Figure 7 shows partial level schemes of the two $\nu$ isotopes, $^{19}$F and $^{138}$La, and two p nuclei, $^{164}$Er and $^{190}$Pt. The half-lives of the first and second excited states in $^{19}$F are 581 ps and 89.3 ns, respectively. $^{138}$La is an important $\nu$-isotope, but the half-lives of its excited states are not known. The half-lives of the first excited states in $^{164}$Er and $^{190}$Pt are 1.47 ns and 60 ps, respectively. It is well known that the nuclei in the rare-earth region are well deformed. As a result, the electric quadrupole transition is enhanced by the collective rotational motion of the deformed nuclei, and the half-lives of their excited states become relatively short. When the plasma generated in $^{19}$F or $^{190}$Pt target can be kept longer than 1 ns, the first excited state in $^{19}$F or $^{190}$Pt can be populated.

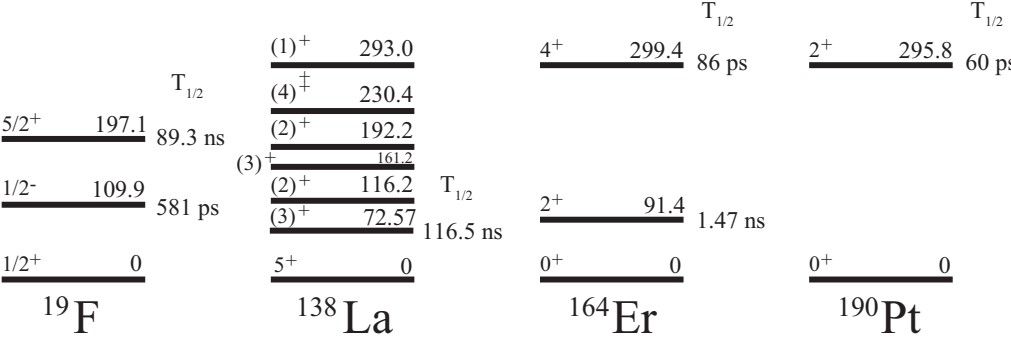

**Figure 7.** Partial level schemes of $^{19}$F, $^{138}$La, $^{164}$Er, and $^{190}$Pt. The states with excitation energies lower than 300 keV are presented. Time indicates the half-lives of excited states.

To measure the cross-section on excited states, Hayakawa and Nakamura proposed nuclear experiments using the following procedure [31]. A laser pulse is divided into a pre-pulse to create hot plasma and a main pulse to generate a high intensity quantum pulse. The time difference between two pulses should be shorter than the lifetime of the plasma produced by the pre-pulse. The pre-pulse irradiates the main target, producing plasma inside the target. If the lifetime of plasma is enough longer than the half-life of an excited state, the population of the excited state is in thermal equilibrium following Equation (4). The main pulse induces the $\gamma$-ray generation target to produce the high flux $\gamma$-ray pulse via laser–plasma interactions. These $\gamma$ rays subsequently irradiate the excited nuclei in the hot plasma before the disappearance of the plasma. Note that the technique wherein the primary laser

pulse is divided to two pulses has been widely used to evaluate physical parameters of hot plasmas generated by high peak power laser (for example, see [32]).

An important consideration in this type of experiments is the relationship between the lifetime of plasma generated by the first pre-pulse and the half-life of an excited state. We estimate the lifetime of plasma using a simple assumption. Let us consider that a laser pulse with an energy of $E_l = 10$ J irradiates a $^{19}$F target. If all of the energy of the laser pulse is absorbed in a target to generate plasma with an average energy of $E_{ph} = 100$ keV and electron density of $n_e = 10^{24}$ cm$^{-3}$, the size of the plasma is $L = 8.5$ μm from an equation of $E_l = n_e E_{ph} L^3$. In the case of stellar plasma, the gravity balances with the plasma pressure, whereas in the laser plasma, the electrostatic pressure of the plasma keeps the plasma against the plasma pressure at first, but the plasma pressure finally leads to the Coulomb explosion. The plasma pressure $P$ is $1.6 \times 10^{16}$ Pa from $P = n_e kT$, and the electrostatic pressure of the plasma is $E = 0.7 \times 10^{14}$ V/m from $P = \varepsilon_0 E^2/2$. The lifetime of the plasma can be estimated to be approximately 170 fs by using the equation of $T = \sqrt{2LM_{atom}/(eE)}$, where $M_{atom}$ is the mass of the atom. The plasma lifetime is shorter than typical half-lives of excited states in nuclei of astrophysical interest, as shown in Figure 7. This result shows that the two-pulse method [31] using laser with a pulse energy of 10 J is not effective for this type of experiment. This estimation indicates that the high power laser which has been developed to study laser nuclear fusions such as NIF [33] and LFEX [34] is required for the generation of pre-pulse. For example, the laser of the National Ignition Facility has an energy of 2 MJ and a pulse duration of 4 ns. Note that the pulse duration is particularly important for simulation of the stellar environment, as discussed above.

### 3.4. Measurements of Isomer Transition Probability

Isomers such as $^{26m}$Al, $^{176m}$Lu, $^{186m}$Re, and $^{180m}$Ta sometimes play an important role in nucleosyntheses. They may be branching points in nucleosynthesis flows. The isomer ratio at the end of a nucleosynthesis process may work as nuclear thermometer, from which the temperature in the stellar environment where the nucleosynthesis process carried out can be estimated. If the excitation energy of a state populated by the absorption of a $\gamma$ ray is lower than the particle threshold, the excited state should decay to the ground state or an isomer by the emission of a $\gamma$ ray or a cascade of $\gamma$ rays (see Figure 6). As a result, in high-temperature environments, the ground state and an isomer are linked by $(\gamma, \gamma')$ reactions through highly excited intermediate states (IMSs) at approximately 1 MeV. Moreover, the transition rate between the ground state and the isomer is affected by the changing temperature. In supernova explosions, the temperature in outer layers decreases on the time-scale of seconds. Therefore, the final isomeric branching ratio should be evaluated by a time-dependent calculation [35]. The excitation energy of IMSs is typically lower than the neutron threshold by an order of magnitude, and the transitions between the ground state and the isomer are more strongly affected at low temperature.

An unstable isotope of $^{26}$Al ($T_{1/2} = 7.2 \times 10^5$ y. is known as a $\gamma$-ray emitter in the Milky Way galaxy. The $\gamma$-rays of $^{26}$Al were measured by $\gamma$-ray detectors in satellites and the frequency of the supernova explosions—which are considered to be one of the main sources of $^{26}$Al in our galaxy—was estimated [36]. There is an isomer with a half-life of 6.2 s in $^{26}$Al. This isomer decays to the ground state of a daughter nucleus $^{26}$Mg without the emission of $\gamma$-rays. Thus, the population ratio of the isomer in high-temperature environments is critical for the estimation of the frequency of the supernovae in our galaxy as well as the origin of $^{26}$Al.

The isomer $^{180}$Ta is meta-stable ($T_{1/2} > 10^{15}$ y.), whereas the ground state decays out with a half-life of 8.15 h. The isomer synthesized by any nucleosynthesis processes may decay to the ground state through intermediate states located above the isomer via $(\gamma, \gamma')$ reactions at high temperatures of $(0.1–1) \times 10^9$ K [35,37]. The residual abundance of $^{180}$Ta is therefore affected by the transition probability between the ground state and the isomer.

The measurement of the transition probability is important for the study of the effect of the isomers. The method to directly measure stellar nuclear reaction rate can be applied to measure the isomer transition rate. Note that the isomer production ratios in ($\gamma$, n) reactions are also important.

## 4. Conclusions

It is possible to generate $\gamma$-ray pulse by the interaction between materials and a high peak power laser. These $\gamma$-ray pulses have the excellent features of ultra-short pulse, high intensity , and continuous energy distribution. These features are suitable for the study of nuclear photoreactions in $\gamma$ and $\nu$ processes. We propose the direct measure of stellar reaction rates with laser-driven $\gamma$-pulse with qusai-stellar spectrum, which can be generated by various laser–plasma interactions. We demonstrated that the bremsstrahlung $\gamma$-rays generated by tuned continuous energy electrons accelerated by laser-induced plasma may have a quasi-stellar energy spectrum. This method can be applied to measure the isomer production ratio and the stellar transition probability between the ground state and an isomer. To measure the reaction rate on excited states, we propose two laser pulse methods in which nuclei excited in the plasma generated by the pre-pulse are irradiated by the quantum beam generated by the main laser pulse.

**Acknowledgments:** This work was supported by JSPS KAKENHI Grant Number 25400540 and 16K13728.

**Author Contributions:** T.H. wrote the main text. T.N. calculated laser driven quantum beam using a PIC simulation code. H.K. suggested the concept of the two pulse experimental procedure. M.K. contributed the details of laser and laser experiments. T.K. contributed to the discussion for nuclear astrophysics.

**Conflicts of Interest:** The authors declare no conflict of interest. The founding sponsors had no role in the design of the study; in the collection, analyses, or interpretation of data; in the writing of the manuscript, and in the decision to publish the results.

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
