# Peer review of "Explosive Nucleosynthesis Study Using Laser Driven γ-ray Pulses"

_qubs, doi:10.3390/qubs1010003_

Round 1

Reviewer 1 Report

The paper discusses possible experiments to measure cross sections relevant to nuclear astrophysics using laser generated gamma rays. After reading the paper I confess I was not able to understand what the novelty is. I think the authors should put more effort in explaining what is the goal of their research. In doing that, they should keep in mind a few factors. As discussed in the introduction, several laboratories are being constructed (or have been constructed) exactly to do investigations in the spirit of this paper. For instance ELI-NP will be providing gamma rays in the region of interest for nuclear astrophysics. Some of the reactions discussed here can be found in the ELI-NP white books/proposal for first experiments. If the goal of the author is the use of a continuous gamma ray distribution to measure the cross section, also this has been discussed in the literature even though for different observables,    M. Barbui, et al., Phys. Rev. Lett. 111, 082502 (2013). Gamma ray measurements are already available, see for instance Henderson et al. High Energy Density Physics 12 (2014) 46.

The authors should put more effort in explaining the novelty of their proposal before the paper can be published. Furthermore they should include some of the available literature. I suggested some papers above, but there are more available.

Author Response

Reviewer #1

Reviewer’s commnet:

The paper discusses possible experiments to measure cross sections relevant to nuclear astrophysics using laser generated gamma rays. After reading the paper I confess I was not able to understand what the novelty is. I think the authors should put more effort in explaining what is the goal of their research. In doing that, they should keep in mind a few factors. As discussed in the introduction, several laboratories are being constructed (or have been constructed) exactly to do investigations in the spirit of this paper. For instance ELI-NP will be providing gamma rays in the region of interest for nuclear astrophysics. Some of the reactions discussed here can be found in the ELI-NP white books/proposal for first experiments. If the goal of the author is the use of a continuous gamma ray distribution to measure the cross section, also this has been discussed in the literature even though for different observables, M. Barbui, et al., Phys. Rev. Lett. 111, 082502 (2013). Gamma ray measurements are already available, see for instance Henderson et al. High Energy Density Physics 12 (2014) 46.

Response to Reviewer’s comment:

    First, we would like to thank the reviewer #1 for his/her kind and suggestive comments. We did not know the previous paper (M. Barbui, et al., Phys. Rev. Lett. 111, 082502 (2013)). We found that this was just a pioneering study for the Big-Bang nucleosynthesis using a high peak power laser. Unfortunately, it has not clearly been written an important fact that this study can contribute to the understanding the Big-Bang nucleosynthesis and the measured energy has been still low. We added this work in the introduction. We also added the reference for the measurement of gamma-ray energy spectra generated the Texas Peta-watt Laser. We advanced our previous manuscript to present our aim and importance. We again thank for the information for the previous papers.

Reviewer 2 Report

The paper under review presents interesting physics, and the authors are

well-known experts in their fields. Whereas the new physical ideas clearly

deserve publication, the manuscript requires some fine-tuning. On the one

hand, the paper is difficult to read because the authors jump back and forth

between different subjects. On the other hand, there are several minor

inaccuracies which should be corrected.

Let me first provide some examples for "jumping back and forth". 

1) Sect.2 "Results" presents spectra of laser/plasma produced

gamma-rays. Sect.3 "Discussion" discusses mainly the astrophysical

relevance. Sect.4 "Materials and Methods" comes back to the production of

laser/plasma gamma-rays.

I suggest to combine Sect. 2 and 4. The headlines should better reflect the

following content.

2) Sect.3.2 presents the measurement of photon-induced (g,n) reaction cross

sections. Sect.3.3 moves to the role of isomers. Sect.3.4 comes back to (g,n)

reactions and takes into account the thermal excitations of the target under

stellar conditions (which is a very essential point, see also 8) below).

Here I suggest to exchange Sect.3.3 and 3.4.

3) In Sect.3.1 "Role of photons..." the authors first discuss the importance

of (g,n) reactions in the gamma-process. Then they change (not in line with

the headline of this section) to neutrino-induced nucleosynthesis (l.64) and

mention the examples 7Li, 11B, 19F, 138La, and 180Ta (l.68). In the next lines

they discuss 92Nb which is not yet mentioned in the above list of relevant

isotopes. Finally, in l.79 they come back to 138La.

Furthermore, the following points should be considered by the authors:

4) p.2, l.43/44:

The authors provide a "Planck distribution of the peak energy of 225

keV". Do the authors mean here that the maximum of Eq.(1) is located at 225

keV? The authors should also provide the usual quantity kT for this

distribution. Furthermore, temperatures of 10^8 - 10^9 K (last sentence)

correspond to kT = 8.6 to 86 keV, i.e. below the shown 225 keV spectrum.

5) p.2, l.56:

The authors state that 27 p-nuclei were synthesized by the

gamma-process. Traditionally, 35 nuclei are named "p-nuclei". This should at

least be mentioned here. Indeed, there has been some discussion on the

one-to-one relation beteeen "p-nucleus" and "p-process/gamma-process" and on

contributions of other nucleosynthetic processes to some p-nuclei. I suggest

to add a reference to [A] here.

6) The discussion of 92Nb should include references to recent work [B,C,D].

7) p.3, l.100:

The quasi-stellar neutron spectrum from the 7Li(p,n) reaction provides kT = 25

keV only. The 3H(p,n) reaction is used for kT=52 keV [E], and the 18O(p,n)

reaction has been suggested for kT = 5 keV [F].

8) p.3, l.108:

At the end of this paragraph I miss a clear statement that the reaction rate

in Eq.(2) is a ground state rate as measured in the laboratory which requires

siginificant theoretical corrections which my reach 3 orders of magnitude

[G]. Note that this huge correction strengthens the need for experiments as

suggested in Sect.3.4!

9) p.4, l.123:

Here I miss a general introductory statement that the isomer

population/depopulation typically proceeds via IMS at excitation energies of

about 1 MeV, i.e. a factor of 10 lower than the energy required for (g,n)

reactions. It should also be stated that under the high temperatures of

explosive scenarios thermal equilibrium is achieved, and that the transition

rates define the freeze-out temperature, see [18,19].

10) p.5, Eq.(5):

The consequences of Eq.(5) for Fig.5 should be discussed. Interestingly, for a

thermally excited state at e.g. 1 MeV, the required energy for the (g,n)

reaction is reduced by exactly this 1 MeV. For all excitation energies this

leads to the same Gamow-like energy window in terms of the excitation energy

of the target nucleus.

11) p.5, l.158:

The "mean-lives" are half-lives. Mean lifetimes and half-lives differ by a

factor of ln(2).

12) p.5, l.175++:

Either the authors intentionally do a very coarse rounding here (which should

be mentioned), or the given numbers are inaccurate. E.g., n_e * E_p * L^3 =

10^24 cm^-3 * 1.6e-14 J * (10^-3 cm)^3 = 16 J for L = 10 \mu m. The plasma

pressure should read P = n_e * kT (l.183). "presser" should probably read

"pressure".

13) p.6, l.208:

"temperature of 300 eV" should probably read "300 keV".

14) P.7, Fig.2: 

Is it possible to include the primary electron spectrum? What is the reason

for the scatter of the black curve at low energies? Statistics?  X-rays? From

the black line, I read a broad maximum around 200-300 keV whereas the authors

state a peak energy of 10-100 keV. (Indeed, there is a local and narrow

maximum between 60-70 keV.)

15) p.9, Fig.5:

This is an almost identical copy of Fig.1 of [14] and Fig.2 of [G]. I suggest

to provide data for a particular nucleus (e.g., one of the cerium isotopes

which are mentioned in the text). This avoids potential conflicts with

copyright and plagiarism.

The solid line does not indicate the reaction rate, but the integrand of the

reaction rate in Eq.(5).

16) p.9, Fig.7:

The source of the data should be specified. The first excited state in 19F is

definitely a 1/2- (not 1/2+), and the 293 keV state in 138La is assigned with

(1)+ in ENSDF instead of (0)+. The caption misses 190Pt, and for clarity it

should be stated that the given times are half-lives.

17) Throughout the paper, the correct name "Planck" is sometimes misspelled as

"Plank".

Further references:

[A] Dillmann, PoS(NIC X)091

[B] Lugaro, PNAS 113, 207 (2016)

[C] Pignatari, IJMPE 25, 1630003 (2016)

[D] Mohr, PRC 93, 065804 (2016)

[E] K"appeler, PRC 35, 936 (1987)

[F] Heil, PRC 71, 025803 (2005)

[G] Vogt, PRC 63, 055802 (2001); follow-up of [14].

Author Response

Reviewer #2

Reviewer’s comment:

The paper under review presents interesting physics, and the authors are

well-known experts in their fields. Whereas the new physical ideas clearly

deserve publication, the manuscript requires some fine-tuning. On the one

hand, the paper is difficult to read because the authors jump back and forth

between different subjects. On the other hand, there are several minor

inaccuracies which should be corrected.

Let me first provide some examples for "jumping back and forth". 

Response to Reviewer’s comment:

   We thank the reviewer #2 for his/her professional and many suggestions. The quantum beam science is a new journal and there is no published sample. Thus, it is difficult to prepare our manuscript to justify the demand format. They require us to present, after the introduction, the result.

Reviewer’s comment:

1) Sect.2 "Results" presents spectra of laser/plasma produced

gamma-rays. Sect.3 "Discussion" discusses mainly the astrophysical

relevance. Sect.4 "Materials and Methods" comes back to the production of

laser/plasma gamma-rays.

I suggest to combine Sect. 2 and 4. The headlines should better reflect the

following content.

Response to Reviewer’s comment:

We agree with the reviewer. We moved the old Sect. 4 to before Sect. 2. And, the section title was changed to “Calculation method and Result.”

Reviewer’s comment:

2) Sect.3.2 presents the measurement of photon-induced (g,n) reaction cross

sections. Sect.3.3 moves to the role of isomers. Sect.3.4 comes back to (g,n)

reactions and takes into account the thermal excitations of the target under

stellar conditions (which is a very essential point, see also 8) below).

Here I suggest to exchange Sect.3.3 and 3.4.

Response to Reviewer’s comment:

We agree with the reviewer, and we exchanged the two sub-sections.

Reviewer’s comment:

3) In Sect.3.1 "Role of photons..." the authors first discuss the importance

of (g,n) reactions in the gamma-process. Then they change (not in line with

the headline of this section) to neutrino-induced nucleosynthesis (l.64) and

mention the examples 7Li, 11B, 19F, 138La, and 180Ta (l.68). In the next lines

they discuss 92Nb which is not yet mentioned in the above list of relevant

isotopes. Finally, in l.79 they come back to 138La.

Response to Reviewer’s comment:

The early studies by Woosley, 1990 and Herger, 2005 did not study for 92Nb, and Hayakawa et al. (2013) proposed the neutrino origin for 92Nb for the first time. Thus we do not add 92Nb in the list. To make it clear, we changed the structure of the introduction as the revised manuscript.

Furthermore, the following points should be considered by the authors:

Reviewer’s comment:

4) p.2, l.43/44:

The authors provide a "Planck distribution of the peak energy of 225

keV". Do the authors mean here that the maximum of Eq.(1) is located at 225

keV? The authors should also provide the usual quantity kT for this

distribution. Furthermore, temperatures of 10^8 - 10^9 K (last sentence)

correspond to kT = 8.6 to 86 keV, i.e. below the shown 225 keV spectrum.

Response to Reviewer’s comment:

The “peak energy” was wrong, and thus we changed it to as “kT=225 keV”. The energy range is also changed to 10^8-3x10^9 K to cover typical peak temperature in gamma-process layers in supernovae.

Reviewer’s comment:

5) p.2, l.56:

The authors state that 27 p-nuclei were synthesized by the

gamma-process. Traditionally, 35 nuclei are named "p-nuclei". This should at

least be mentioned here. Indeed, there has been some discussion on the

one-to-one relation beteeen "p-nucleus" and "p-process/gamma-process" and on

contributions of other nucleosynthetic processes to some p-nuclei. I suggest

to add a reference to [A] here.

Response to Reviewer’s comment:

We added such sentences. Hayakawa (2008) discussed that 35 p-nuclei can be classified into 4 groups and among 35 p-nuclei, only 27 p-nuclei can be produced by the gamma-process. We added the ref [A] into the reference [Hayakawa, 2008].

Reviewer’s comment:

6) The discussion of 92Nb should include references to recent work [B,C,D].

Response to Reviewer’s comment:

In our paper, the aim for the 92Nb discussion is not to discuss the origin of 92Nb but to present the fact that the study of the neutrino-process has been active now. However, we added a sentence “Note that the origin of $^{92}$Nb has been studied still [B,D].”

Reviewer’s comment:

7) p.3, l.100:

The quasi-stellar neutron spectrum from the 7Li(p,n) reaction provides kT = 25

keV only. The 3H(p,n) reaction is used for kT=52 keV [E], and the 18O(p,n)

reaction has been suggested for kT = 5 keV [F].

Response to Reviewer’s comment:

Yes, we added these two references. Note that we found a recent paper to study 23Na using 18O(p, n) reaction (E. Uberseder, PRC, 2017) and we can say that 18O(p,n) reaction is used for the s-process study now.

Reviewer’s comment:

8) p.3, l.108:

At the end of this paragraph I miss a clear statement that the reaction rate

in Eq.(2) is a ground state rate as measured in the laboratory which requires

siginificant theoretical corrections which my reach 3 orders of magnitude

[G]. Note that this huge correction strengthens the need for experiments as

suggested in Sect.3.4!

Response to Reviewer’s comment:

We agree with the reviewer. We added such sentences with the reference [G].

Reviewer’s comment:

9) p.4, l.123:

Here I miss a general introductory statement that the isomer

population/depopulation typically proceeds via IMS at excitation energies of

about 1 MeV, i.e. a factor of 10 lower than the energy required for (g,n)

reactions. It should also be stated that under the high temperatures of

explosive scenarios thermal equilibrium is achieved, and that the transition

rates define the freeze-out temperature, see [18,19].

Response to Reviewer’s comment:

Hayakawa (2010) showed, in principle, the isomer ratio should be obtained using time-dependent calculation. In specific cases, the sudden freeze-out model may be a good approximation. We added the sentences to explain it.

Reviewer’s comment:

10) p.5, Eq.(5):

The consequences of Eq.(5) for Fig.5 should be discussed. Interestingly, for a

thermally excited state at e.g. 1 MeV, the required energy for the (g,n)

reaction is reduced by exactly this 1 MeV. For all excitation energies this

leads to the same Gamow-like energy window in terms of the excitation energy

of the target nucleus.

Response to Reviewer’s comment:

We agree with the reviewer. We added such sentences.

Reviewer’s comment:

11) p.5, l.158:

The "mean-lives" are half-lives. Mean lifetimes and half-lives differ by a

factor of ln(2).

Response to Reviewer’s comment:

Yes.

Reviewer’s comment:

12) p.5, l.175++:

Either the authors intentionally do a very coarse rounding here (which should

be mentioned), or the given numbers are inaccurate. E.g., n_e * E_p * L^3 =

10^24 cm^-3 * 1.6e-14 J * (10^-3 cm)^3 = 16 J for L = 10 \mu m. The plasma

pressure should read P = n_e * kT (l.183). "presser" should probably read

"pressure".

Response to Reviewer’s comment:

We changed to correct numbers. The result is almost same.

Reviewer’s comment:

13) p.6, l.208:

"temperature of 300 eV" should probably read "300 keV".

Response to Reviewer’s comment:

Yes.

Reviewer’s comment:

14) P.7, Fig.2: 

Is it possible to include the primary electron spectrum? What is the reason

for the scatter of the black curve at low energies? Statistics?  X-rays? From

the black line, I read a broad maximum around 200-300 keV whereas the authors

state a peak energy of 10-100 keV. (Indeed, there is a local and narrow

maximum between 60-70 keV.)

Response to Reviewer’s comment:

Yes, we added the primary electron spectrum. The scatter originated from the statistics. Because the purpose of this figure is to present the similarity between the laser driven gamma-ray spectrum and Planck distribution, we deleted the explanation for the peak energy, which may lead misunderstanding by readers.

Reviewer’s comment:

15) p.9, Fig.5:

This is an almost identical copy of Fig.1 of [14] and Fig.2 of [G]. I suggest

to provide data for a particular nucleus (e.g., one of the cerium isotopes

which are mentioned in the text). This avoids potential conflicts with

copyright and plagiarism.

The solid line does not indicate the reaction rate, but the integrand of the

reaction rate in Eq.(5).

Response to Reviewer’s comment:

We think this figure is the important and common concept for gamma-process. Thus, we changed to a simpler picture and refer to the original figure in Mohr 2001. We also changed the figure caption to avoid the confusing for the peak energy.

Reviewer’s comment:

16) p.9, Fig.7:

The source of the data should be specified. The first excited state in 19F is

definitely a 1/2- (not 1/2+), and the 293 keV state in 138La is assigned with

(1)+ in ENSDF instead of (0)+. The caption misses 190Pt, and for clarity it

should be stated that the given times are half-lives.

Response to Reviewer’s comment:

Thanks.

Reviewer’s comment:

17) Throughout the paper, the correct name "Planck" is sometimes misspelled as

"Plank".

Response to Reviewer’s comment:

Finally, we again thank the reviewer #2 for his/her suggestive comments to improve our manuscript. The revised manuscript is clearly improved.

Round 2

Reviewer 1 Report

The paper has been improved. Accept for publication.

Reviewer 2 Report

The authors have improved their manuscript. There are some technical details in the presentation which should be checked before publication.

1) Several numbers of figures in the text are inconsistent (p.4, l.80: Fig.5 should read Fig.3; l90, l.93: Fig.6 should read Fig.4; etc. etc.)

2) p.3, l.71 mentions a dashed line in Fig.2. There are several colored lines in Fig.2, but no dashed line. Maybe, also the word "photon" should be removed in the x-axis and y-axis labels (as the authors have now - following my suggestion - included the electron spectrum).

3) "Plank" should read "Planck" in Fig.5.